# COVID-19 pandemic restrictions continuously impact on physical activity in adults with cystic fibrosis

Thomas Radtke[1‡]*, Sarah R. Haile[2‡], Holger Dressel[1], Christian Benden[3]

1 Division of Occupational and Environmental Medicine, Epidemiology, Biostatistics and Prevention Institute, University of Zurich and University Hospital Zurich, Zurich, Switzerland, 2 Division of Chronic Disease Epidemiology, Epidemiology, Biostatistics and Prevention Institute, University of Zurich, Zurich, Switzerland, 3 Faculty of Medicine, University of Zurich, Zurich, Switzerland

‡ These authors share first authorship on this work.
* thomas.radtke@uzh.ch

## Abstract

### Background

We have recently reported reduced physical activity (PA) in people with cystic fibrosis (pwCF) with and without lung transplantation (LTX) during a 6-week stringent lockdown in Switzerland. This follow-up study explores the impact of coronavirus-2019 disease (COVID-19) related pandemic restrictions on individuals' therapy regimens and health-related aspects in pwCF.

### Methods

We conducted a cross-sectional web-based national survey in Spring 2021. The survey included questions on daily PA, airway clearance and inhalation therapy, questions on COVID-19-compatible symptoms, diagnostic tests and vaccination status, and enquired health-related aspects covering the pandemic period between March 2020 to April 2021.

### Results

193 individuals with CF (53% female; 25% LTX recipients) participated. Among pwCF, 10 reported COVID-19 (n = 2 LTX recipients), two subjects were hospitalized, no invasive ventilation required, no deaths. The clinical course was generally mild. Overall, 46% reported less PA during the pandemic, mostly due to closed fitness facilities (85%), lack of motivation (34%), and changes in daily structures (21%). In contrast, 32/193 (17%) pwCF were able to increase their PA levels: 12 (38%) and 11 (34%) reported undertaking home-based training and outdoor activities more frequently; 6 (19%) reported an increase in routine PA, and another 3 (9%) started new activities. Among pwCF without LTX, 5% and 4% reported to undertake less airway clearance and inhalation therapy, respectively.

### Conclusions

Our study reveals unfavorable consequences of COVID-19 pandemic restrictions on PA of pwCF with unknown long-term consequences for their overall physical fitness and lung

**Data Availability Statement:** All relevant data are within the manuscript and its Supporting Information files.

**Funding:** The authors received no specific funding for this work.

**Competing interests:** The authors have declared that no competing interest exist.

health. Strategies to overcome this undesirable situation are needed; increased uptake of telehealth PA programs and virtual exercise classes to promote PA participation might be one promising approach along with vaccination of pwCF and their close contacts.

## 1. Introduction

Severe acute respiratory syndrome–coronavirus-2 (SARS-CoV-2) became apparent at the end of 2019 and continuously causes a global pandemic affecting the health and well-being of the general population and those with chronic respiratory disease [1, 2]. According to the Centers for Disease Control and Prevention [3], people with chronic respiratory diseases such as cystic fibrosis (CF) with and without lung transplantation (LTX) are at a higher risk for severe coronavirus-disease 2019 (COVID-19); however, the clinical course and recovery of people with CF (pwCF) after COVID-19 and the potential protective role of CF transmembrane conductance regulator (CFTR) therapy remains to be investigated [4].

Reports from the European Cystic Fibrosis Society Patient Registry and Cystic Fibrosis Registry Global Data Harmonization Group aiming to characterize SARS-CoV-2 infections among pwCF show an overall mild course of COVID-19 [5–8]. LTX recipients and those with pre-existing advanced CF lung disease (ACFLD) and other extra-pulmonary comorbidities seem to be at higher risk for severe COVID-19; even fatal outcomes were reported in a few cases [5, 7, 8]. Interestingly, the incidence of SARS-CoV-2 in pwCF was reported to be higher compared to an age-matched general population [7].

In many countries, people have been advised to strictly limit social contacts and self-isolation has been recommended for pwCF to minimize the risk of SARS-CoV-2 infection. Public restrictions (e.g., closed training facilities and physiotherapy practices, omission of supervised therapy) are likely to have important implications on individuals' therapy regimens, adherence to treatment, and mental health of pwCF [1, 9]. Our Group recently reported lower physical activity levels in a substantial proportion of adults with CF during the first wave of the pandemic including a 6-week nationwide lockdown in Switzerland [10], supported by observational data of pwCF living in the UK [9], and in Greece [11].

While most of the studies to date have reported on the impact of short periods of shielding (i.e., 3 weeks to 4 months) on treatment adherence in pwCF [1, 9–11], the long-term consequences are yet to be determined. The aim of our follow-up survey–covering the pandemic for over one year (from March 2020 to April 2021)–was to assess the continuous impact of SARS-CoV-2 on individuals' therapy regimes, perceived health status and physical activity behavior.

## 2. Materials and methods

This is a follow-up cross-sectional study of our previous survey conducted during the first wave of the SARS-CoV-2 pandemic of pwCF living in Switzerland [10]. The self-administered online questionnaire was programmed in REDCap (Research Electronic Data Capture, Vanderbilt University, USA) and contained questions on demographics and medical characteristics, current therapies, SARS-CoV-2 vaccination status, and individuals' perceptions towards various medical and non-medical aspects related to the SARS-CoV-2 pandemic. In addition, we assessed self-reported health status with the "feeling thermometer", a visual analogue scale (VAS) with marked intervals from 0 (worst imaginable health state) to 100 (best imaginable health state). The feeling thermometer is part of the EuroQoL quality of life questionnaire

(EQ-5D-5L) [12], a valid and frequently applied health outcome instrument in CF lung disease including lung transplant recipients [13–15].

The online questionnaire was pilot tested by three individuals with CF who also provided feedback and input on the first survey [10]. The questionnaire was made available in the three main national languages in Switzerland (French, German, Italian). The original study questionnaires were translated into English (S1 File). According to the latest European Cystic Fibrosis Society Patient Registry Report [16], 539 adults with CF live in Switzerland and belong to the pool of potentially eligibly survey participants.

We classified the severity of COVID-19 into the following stages: Stage I–- mild symptoms (i.e. dry cough, fatigue, headache); Stage II–moderate to severe symptoms (i.e., dyspnea, hypoxia); Stage III–critical symptoms (i.e., acute respiratory distress syndrome, cardiac failure); Stage IV–death [17].

On March 2nd, 2021, one day after easing public restrictions following the second wave of the pandemic in Switzerland (e.g., opening of retail shops, private outdoor meetings with maximal 15 persons, young people <20 years attending sports and cultural events), the online link to the REDCap questionnaire was distributed via email to all current members of Cystic Fibrosis Switzerland. In addition, the link was made available on the official Facebook® page of Cystic Fibrosis Switzerland and shared in WhatsApp® groups by pwCF. The Study Team additionally shared the link with the LTX and CF Center in Lausanne, and other centers caring for pwCF across Switzerland. A second survey invitation was sent as a reminder on March 16th, 2021. The surveys included questions covering the entire SARS-CoV-2 pandemic period from March 2020 to April 2021.

### 2.1 Ethics statement

This study does not fall under the scope of the Human Research Act. Research projects with anonymously collected or anonymous health-related personal data are exempt from requirement for approval. We paid special attention to the anonymity of survey participants, e.g., by using broad categories for age (i.e., 18–24 years, 25–39 years, 40–60 years, and >60 years) and time of LTX (i.e., < 1 year, 1–5 years, >5–10 years, > 10 years). Written consent for participation in this survey was systematically collected at the start of the online questionnaire.

### 2.2 Statistical methods

Descriptive characteristics are summarized as n (%) or median (interquartile range). Data cleaning and analysis was performed using R (version 4.0), and graphs constructed using ggplot2 (version 3.3) [18, 19].

## 3. Results

During the 4-week study period, 193 survey responses were received from pwCF. Overall, 127 (66%) of pwCF who participated in our first survey also participated in the follow-up survey.

### 3.1 Survey participants

Clinical and socioeconomic characteristics of pwCF are given in **Table 1**. Among 193 pwCF, 25% were LTX recipients, of which 3 (6%) reported to have chronic lung allograft dysfunction. Among 144 pwCF without LTX, 62% reported taking CFTR modulators, of which 42% received triple combination therapy (elexacaftor-ivacaftor-tezacaftor). Participants' self-reported health status was different between lung disease categories: mild lung disease (FEV$_1$ >80% predicted) median 84.5 (IQR 73.5, 92.3), moderate lung disease (FEV$_1$ 40–80%

**Table 1. Characteristics of pwCF participating.**

| Variables | All (n = 193) | Non-LTX (n = 144) | LTX (n = 49) |
|---|---|---|---|
| **Sex** | | | |
| Male | 90 (47%) | 69 (48%) | 21 (43%) |
| Female | 103 (53%) | 75 (52%) | 28 (57%) |
| **Age categories** | | | |
| 18–24 years | 39 (20%) | 34 (24%) | 5 (10%) |
| 25–39 years | 101 (52%) | 77 (53%) | 24 (49%) |
| 40–60 years | 49 (25%) | 29 (20%) | 20 (41%) |
| >60 years | 4 (2%) | 4 (3%) | 0 (0%) |
| **Lung function (FEV$_1$)*** | | | |
| > 80% predicted | | 36 (25%) | – |
| 40–80% predicted | | 84 (58%) | – |
| < 40% predicted | | 24 (17%) | – |
| **CFTR modulator therapy** | | | |
| IVA | | 3 (2%) | |
| IVA/LUM | | 7 (5%) | |
| IVA/TEZ | | 19 (13%) | |
| ELX/IVA/TEZ | | 60 (42%) | |
| **Years post LTX** | | | |
| < 1 year | | – | 1 (2%) |
| 1–5 years | | – | 16 (33%) |
| >5–10 years | | – | 20 (41%) |
| > 10 years | | – | 12 (24%) |
| **Comorbidities** | | | |
| Heart disease | 1 (1%) | 1 (1%) | 0 (0%) |
| Hypertension | 24 (12%) | 5 (4%) | 19 (39%) |
| Diabetes | 87 (45%) | 46 (32%) | 41 (84%) |
| Kidney disease/renal transplant | 10 (5%) | 0 (0%) | 10 (20%) |
| Cancer[#] | 2 (1%) | 0 (0%) | 2 (4%) |

Data are presented as numbers (percent). CFTR, cystic fibrosis transmembrane conductance regulator; ELX, elexacaftor; FEV$_1$, forced expiratory volume in 1s; IVA, ivacaftor; LUM, lumacaftor; LTX, lung transplantation; TEZ, tezacaftor.

*Lung function was only surveyed in non-transplant patients.

[#]Other than skin cancer.

predicted) 74.0 (65.0, 83.0), and advanced lung disease (FEV$_1$ <40% predicted) 70.0 (54.5, 82.0), p = 0.001. Self-reported health status reported by lung transplant individuals was 80.0 (68.0, 90.0).

## 3.2 Demographics on COVID-19 compatible symptoms, diagnostic testing, COVID-19 disease stages, and vaccination status

During the entire SARS-CoV-2 pandemic period starting in March 2020, 41 individuals with CF (21%) reported COVID-19 compatible symptoms. Among symptomatic subjects, 37 individuals underwent SARS-CoV-2 testing, 10 individuals (27%) were tested positive (**Table 2**). All experienced mild symptoms only—corresponding to COVID-19 Stage I [17], except two individuals who were hospitalized, one individual was admitted to the intensive care unit but with no invasive ventilation required. 80 of 193 pwCF (42%) reported being already vaccinated for COVID-19 (i.e., at least one dose of a SARS-CoV-2 two-doses vaccine regime), with a

**Table 2. COVID-19 like symptoms, diagnostic testing, disease stages, and vaccination status.**

| Variables | All (n = 193) | Non-LTX (n = 144) | LTX (n = 49) |
|---|---|---|---|
| COVID-19 like symptoms | 41 (21%) | 34 (24%) | 7 (14%) |
| Symptomatic tested[§] | 37 (97%) | 30 (97%) | 7 (100%) |
| SARS-CoV-2 positive | 10 (27%) | 8 (27%) | 2 (29%) |
| COVID-19 Stage I* | 8 (80%) | 7 (88%) | 1 (50%) |
| COVID-19 Stage II* | - | - | - |
| COVID-19 Stage III* | 2 (20%) | 1 (13%) | 1 (50%) |
| Hospitalisation | 2 (20%) | 1 (13%) | 1 (50%) |
| Intensive care | 1 (10%) | 1 (13%) | 0 (-) |
| **Vaccination status** | | | |
| Already vaccinated[#] | 80 (42%) | 57 (40%) | 23 (47%) |
| Not vaccinated but willing to | 67 (35%) | 49 (35%) | 18 (37%) |
| Unwilling to be vaccinated | 16 (8%) | 15 (11%) | 1 (2%) |
| Undecided | 27 (14%) | 20 (14%) | 7 (14%) |

Data are presented as numbers (percent). COVID-19, coronavirus disease 2019; SARS-CoV-2, Severe Acute Respiratory Syndrome Coronavirus 2.

* COVID-19 stages were adapted from Sidiqqi & Mehra[17].

# At least one dose of a SARS-CoV-2 two-doses vaccine regime.

§ Percentages are reported for those tested for SARS-CoV-2.

further 35% willing to get the vaccine; 8% reported not being willing to get vaccinated and the remaining 14% were undecided. 59% of survey respondents (n = 114) reported being informed by their CF center about the COVID-19 vaccination.

### 3.3 Impact of the SARS-CoV-2 pandemic on daily physical activity, airway clearance and inhalation therapy

Table 3 summarizes changes in individuals' daily maintenance therapy. During the pandemic, 48%, 5%, and 4% of individuals without LTX reported to undertake less physical activity, airway clearance and inhalation therapy, respectively. Most frequently reported reasons for doing less physical activity were closing of fitness centers (85%), lack of motivation (33%), lack of daily structure (21%), and cancellation of supervised therapy (19%). Among LTX recipients, 41% reported to undertake less physical activity. Among all survey respondents, 32 (17%) were able to increase their physical activity levels, of which 37% and 33% reported to undertake home-based training and outdoor activities more frequently; 22% reported an increase in routine physical activity, and another 8% started new activities.

### 3.4 Impact of COVID-19 pandemic on individuals' health-related and socioeconomic aspects

Perceptions of pwCF on health-related and socioeconomic aspects including concern of worsening of their lung disease, worries about their financial situation, job loss, and social isolation from the first survey and the follow-up survey are given in the online supplement (S2 File).

### 4. Discussion

To the authors' knowledge, this is the first national survey among pwCF addressing health-related aspects related to the COVID-19 pandemic covering a one-year period. Overall, the clinical impact appears mild with 10 (5%) individuals being affected by COVID-19, no fatal

**Table 3. Impact of the SARS-CoV-2 pandemic on individuals' daily physical activity, airway clearance and inhalation therapy.**

| Variables | All | Non-LTX |
|---|---|---|
| **Daily physical activity** | | |
| Total subjects | 193 | 144 |
| No change in therapy | 72 (37%) | 53 (37%) |
| Less frequently | 89 (46%) | 69 (48%) |
| More frequently | 32 (17%) | 22 (15%) |
| **Daily airway clearance** | | |
| Total subjects | 153 | 136 |
| No change in therapy | 141 (92%) | 124 (91%) |
| Less frequently | 7 (5%) | 7 (5%) |
| More frequently | 5 (3%) | 5 (4%) |
| **Daily inhalation therapy** | | |
| Total subjects | 170 | 141 |
| No change in therapy | 162 (95%) | 133 (94%) |
| Less frequently | 5 (3%) | 5 (4%) |
| More frequently | 3 (2%) | 3 (2%) |

Data are presented as numbers (percent). LTX, lung transplantation.

outcome. COVID-19 pandemic related restrictions seem to have a continuous impact on individuals' physical activity levels, a cornerstone therapy in the modern era of CF care, whereas daily airway clearance and inhalation therapy seem less affected.

In our cohort, the clinical impact of the SARS-CoV-2 pandemic was overall mild with 10 individuals reporting of COVID-19. Of those, two individuals (one LTX recipient) were hospitalized, one admitted to the intensive care unit but without need for invasive ventilation. Our data are consistent with recent reports from the Cystic Fibrosis Registry Global Data Harmonization Group and European Cystic Fibrosis Society Patient Registry reporting an overall mild course of COVID-19 in the majority of pwCF [5–8]. In the most recent report of the Cystic Fibrosis Registry Global Data Harmonization Group [8] including 181 pediatric and adult cases with SARS-CoV-2 covering a broad spectrum of CF lung disease severity including LTX recipients from 19 countries, 11 (6%) were admitted to the intensive care unit; 7 (4%) deaths were reported. Additionally, data collected within the European Cystic Fibrosis Society Patient Registry as of May 21 2021, confirms the overall mild clinical course of COVID-19 in pwCF in 1459 cases of which 229 were hospitalized, 33 required intensive care unit admission, 30 needed ventilatory support, and 17 pwCF died [20].

Our survey revealed long-term consequences of COVID-19 shielding on individuals' physical activity behavior with 46% of pwCF reporting to undertake less physical activity during the pandemic. Reduced physical activity levels during earlier phases of the COVID-19 pandemic have also been reported in people living with chronic obstructive pulmonary disease and renal disease [21, 22]. Our data suggest unfavorable and prolonged effects of COVID-19 restrictions on a cornerstone therapy of CF care, given the fact that a similar proportion of pwCF in this survey (i.e., 45% in our previous survey covering a shorter time-period during the first phase of the pandemic) reported less daily physical activity. The beneficial effects of regular physical activity are well reported in pwCF without LTX, showing that physically active individuals–compared to less active individuals–have a slower rate of lung function decline, and reduced hospital admissions [23–25]. This is even more important in the era of modern CF therapy with a substantial proportion of pwCF using physical activity and exercise (i.e., structured

physical activity) as a stand-alone therapy for airway clearance [26, 27]. It is important to note that Switzerland is among the European countries that experienced one of the highest second waves during the SARS-CoV-2 pandemic in Autumn 2020, but public restrictions were comparably modest. Nevertheless, access to fitness centers were limited for several months and was reported as the main reason for doing less physical activity by 85% of our survey participants. Moreover, many pwCF were advised to self-isolate and to limit social contacts, thus reducing abilities for being physically active. Fortunately, 32 (17%) of survey respondents were able to increase their physical activity levels during the pandemic. However, based on participants' characteristics (e.g., age, sex, disease severity, CFTR modulator therapy, vaccination status, employment status and work percentage) we were not able to identify determinants for this positive behavioral change. Among those, about two-thirds chose to undertake home-based training or outdoor activities more frequently, while about one-fifth increased their usual physical activity routine, and the remaining started new activities. Moreover, in our first survey, five pwCF reported to self-organize informal virtual exercise groups. The group still exists, and they reported to meet daily during the weekdays for about 30 minutes to perform endurance type and strengthening exercises together (personal communication to first author). Telehealth interventions and virtual exercise classes to deliver and promote physical activity among pwCF appear feasible [28–30]; such tools could be further developed by healthcare professionals in collaboration with pwCF and upscaled offering supervised home-based exercise training to pwCF during periods when training facilities are closed. Moreover, virtual exercise groups offer an elegant opportunity for pwCF to exercise with their peers, that is usually not possible and not recommended due to risk of cross infection. Finally, a recent large scale analysis of 48'440 adults with COVID-19 revealed that people who are consistently physically inactive have a substantially higher risk for severe COVID-19 compared to people meeting physical activity guidelines (i.e., at least 150 min of moderate-intensity physical activity per week) [31]. To which extend these data could be extrapolated to pwCF remains to be determined; however, no doubt exists that regular physical activity is beneficial to the overall health of pwCF.

With very limited therapy options available to successfully treat COVID-19, preventive measures remain the key to prevent severe COVID-19. Just one year into the pandemic, several COVID-19 vaccines have already been approved by regulatory authorities, e.g., in the European Union, the United Kingdom, and the United States of America (USA). Although vaccines play a major role as a way out of the pandemic, willingness to receive a COVID-19 vaccination varies among people. Governments and health officials generally encourage vulnerable populations to sign up for the COVID-19 vaccination including people with chronic lung diseases such as CF. Further, patient support organizations such as the Cystic Fibrosis Foundation (CFF) in the USA strongly support COVID-19 vaccine recommendations for pwCF [32]. Our survey reveals that more than three-quarters of participating pwCF are either already vaccinated against COVID-19 or willing to do so, no difference among transplanted pwCF and non-transplanted pwCF was detected. In comparison to the general population in Switzerland and Liechtenstein–by April 2 2021–6.7% had been fully vaccinated and 4.5% had received at least one dose of a two-dose vaccination regimen, respectively [33]. To date there is limited understanding of COVID-19 vaccine immunogenicity and safety in solid organ transplant recipients as this at-risk population was excluded from the initial vaccine trials. Further, a recent study by Boyarsky et al. [34] on the immunogenicity of the mRNA SARS-CoV-2 two-dose vaccine regimes among 658 solid organ transplant recipients revealed a measurable anti-spike antibody response in 54% of individuals, whereas 46% had no such response, suggesting that a substantial proportion are still at risk for COVID-19 [34]. Nevertheless, the COVID-19 Taskforce of the International Society for Heart and Lung Transplantation (ISHLT) endorses

vaccination for transplant recipients as the benefits of receiving SARS-CoV-2 vaccination outweigh the risks; ideally, transplant recipients should be included into vaccine trials after consultation with their responsible transplant physician [35]. Furthermore, even fully vaccinated transplant recipients should continue hygiene measures until more data are available on vaccine efficacy post-transplant. In addition, the ISHLT COVID-19 Taskforce recommends vaccination of immunocompetent household contacts of transplant recipients [35].

### 4.1 Limitations

This study has several limitations. First, with 193 survey responses, the participation rate of pwCF in this second survey was lower compared to our previous survey (n = 327, 63% response rate) covering a shorter time-period following the nationwide lockdown in Switzerland. With this second survey, we were only able to reach about 36% of potentially eligible adults with CF living in Switzerland [16]. Nevertheless, the proportion of individuals with LTX was comparable to our previous survey (25%) with no differences across age categories, sex, and lung disease severity between survey respondents potentially affecting the generalizability of our findings to adult pwCF in Switzerland. We used the same information channels to distribute the survey link and the time-period of 4 weeks was identical to our first survey. It may well be that the lower interest in this follow-up survey reflects an overall weariness ("pandemic fatigue") among pwCF in regard to COVID-19 [36]. Second, we assessed self-reported physical activity in simple categories ("more frequently", "less frequently", "unchanged") to gain insights into long-term behavioral changes potentially caused with pandemic restrictions. Objectively measured physical activity would have provided additional information about the magnitude of changes, both on an individual and population-level. However, this would have required a prospective study design and longitudinal measurements of free-living physical activity over time. Thirdly, since we assessed COVID-19 test results based on reported symptoms using branching logics in our online survey, our report may underestimate the number of positive cases due to the fact that a non-negligible proportion of positive SARS-CoV-2 tests is observed among asymptomatic pwCF [5]. Finally, individuals affected by COVID-19 or with a COVID-19 case among family, friends or acquaintances are probably more likely to complete our survey. However, among pwCF, only 21% reported COVID-19 compatible symptoms during the pandemic; thus, it seems rather unlikely that we report on a selective cohort of adults with CF.

### 5. Conclusions

In summary, our study reveals unfavorable consequences of COVID-19 pandemic restrictions on physical activity of pwCF long-term; its potential impact on physical fitness and lung health has yet to be determined. Strategies to overcome this undesirable situation are needed; increased uptake of telehealth physical activity programs and virtual exercise classes to promote physical activity participation might be one promising approach.

### Supporting information

**S1 File. Questionnaire to assess the state of health of people with cystic fibrosis in times of the coronavirus pandemic (COVID-19).**
(DOCX)

**S2 File. Perceptions of individuals with cystic fibrosis on health-related and socioeconomic aspects during the coronavirus pandemic.** Disease categories for non-transplant individuals were based on lung function, i.e., percent predicted forced expiratory volume in 1s ($FEV_1$):

mild CFLD (>80%), moderate CFLD (80% - 40%), and advanced CFLD (<40%). CFLD, cystic fibrosis lung disease; LTX, lung transplantation. Red bars = first survey (n = 327, 83 LTX); Green bars = follow-up survey (n = 193, 49 LTX).
(PDF)

**S3 File. Individual data from all participants.**
(XLSX)

## Acknowledgments

The authors acknowledge the support of Cystic Fibrosis Switzerland (CFS) and its Executive Board Members, Andreas Jung, MD, and Reto Weibel, for distributing the link of the online survey via the e-mail to CFS Members. We further acknowledge Reta Fischer, MD (Quartier Bleu, Berne, Switzerland); Alain Sauty, MD (Hospitalier Neuchâtelois, Neuchâtel, Switzerland); Thomas Geiser, MD (Inselspital, Berne, Switzerland); Jérôme Plojoux, MD (Hôpitaux Universitaires de Genève, Genève, Switzerland), Christian Murer, MD (Kantonsspital Luzern, Lucerne, Switzerland); Sarosh Irani, MD (Kantonsspital Aarau, Aarau, Switzerland); John-David Aubert, MD, and Angela Koutsokera, MD, PhD (Centre Hospitalier Universitaire Vaudois, Lausanne, Switzerland), and Kathleen Jahn, MD (Universitätsspital Basel, Basel, Switzerland) for circulating the online survey link to patients at their respective centers. Moreover, we kindly thank Daniele Marino, MD, and Angela Koutsokera, MD, PhD, for their assistance with the translations of the questionnaires and their critical feedback into Italian and French, respectively. Finally, we acknowledge all individuals with CF for their participation.

## Author Contributions

**Conceptualization:** Thomas Radtke, Christian Benden.

**Formal analysis:** Sarah R. Haile.

**Methodology:** Thomas Radtke, Holger Dressel, Christian Benden.

**Project administration:** Thomas Radtke, Christian Benden.

**Resources:** Holger Dressel.

**Validation:** Thomas Radtke, Sarah R. Haile.

**Visualization:** Sarah R. Haile.

**Writing – original draft:** Thomas Radtke.

**Writing – review & editing:** Sarah R. Haile, Holger Dressel, Christian Benden.

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
