## [Decision Letter · Decision Letter 0]

11 Aug 2021

PONE-D-21-21163

COVID-19 pandemic restrictions have long-term impact on physical activity in adults with cystic fibrosis

PLOS ONE

Dear Dr. Radtke,

Thank you for submitting your manuscript to PLOS ONE. After careful consideration, we feel that it has merit but does not fully meet PLOS ONE’s publication criteria as it currently stands. Therefore, we invite you to submit a revised version of the manuscript that addresses the points raised during the review process.

We look forward to receiving your revised manuscript.

Kind regards,

Sinan Kardeş, M.D.

Academic Editor

PLOS ONE

Journal Requirements:

Reviewers' comments:

Reviewer's Responses to Questions

**Comments to the Author**

1. Is the manuscript technically sound, and do the data support the conclusions?

Reviewer #1: Yes

Reviewer #2: Yes

Reviewer #3: Yes

2. Has the statistical analysis been performed appropriately and rigorously? 

Reviewer #1: Yes

Reviewer #2: Yes

Reviewer #3: Yes

3. Have the authors made all data underlying the findings in their manuscript fully available?

Reviewer #1: Yes

Reviewer #2: Yes

Reviewer #3: Yes

4. Is the manuscript presented in an intelligible fashion and written in standard English?

Reviewer #1: Yes

Reviewer #2: Yes

Reviewer #3: Yes

5. Review Comments to the Author

Reviewer #1: In this paper the authors present the second online survey to explore the impact of pandemic restrictions to combat COVID-19 on pwCF therapeutic regimens and physical well-being.

The authors also included CF patients with lung transplantation. The results confirm that the pandemic negatively affects all patients as a significant number of them stop exercising, and the others are training autonomous and uncontrolled activities, therefore the long-term consequences on the physical and lung health of these patients are not foreseeable.

The work is interesting and confirms the data reported in a previous work by the same authors, and it is important to reiterate that physical activity is a fundamental practice for the anthropometric, clinical, and metabolic improvement of pwCF regardless of the pandemic, as reported in the work of Elce et al, published in Clin Respir J. 2018; 12 2228–2234. Authors should cite this work at the following points in the manuscript:

- page 3, line 67 in addition to the other citations;

- page 13, line 209, change "regular" with "supervised" and "activity" with "exercise";

- page 13, line 211, add after admission, "and improves clinical, anthropometric and metabolic parameters;

- page 16, line 283, after "over time".

Reviewer #2: This is an interesting follow up study regarding physical activity levels and maintenance therapy adherence in CF patients during the COVID-19 pandemic. Even though it received fewer participants’ responses than the original study it was still representative of the population and this limitation was well noted on the manuscript. The authors provided an adequate review of the literature and the manuscript is well organized.

However I would suggest some reviews:

1- In the Introduction section the authors list as one of their objectives to assess the long-term impact of SARS-CoV-2 on individuals’ perceived health status. However, this is not approached in the results or the conclusions. The questionnaire has one question regarding perceived health status (question 15: “We would like to know how you regard your health status today”), but answers are not described in the study. It would be interesting to include this data, and to evaluate whether the change in physical activity were associated with perceived health status.

2- It would also be interesting to investigate whether vaccination status was associated to higher physical activity.

3- In Supplement 2, the colors in the figures are lacking captions.

4- In section 3.3 the reduction of airway clearance and inhalation therapy is noted twice (in line 159 and 166).

Reviewer #3: Thank you for giving me the opportunity to read this very interesting survey by Radtke et al. Overall, the authors should be acknowledged for what it a well-conducted survey and a clear and concise reporting of these valuable results. The COVID-19 pandemic have raised very important questions for the CF community of patients and caregivers, and this survey provides relevant results and appropriate messages.

My main concern is about the terminology employed by the authors concerning the "long-term" effect of the pandemic restrictions. I am not fully comfortable with this wording of "long term consequences" employed by the authors. I fully agree with the interest of this one-year survey but, to my opinion, the pandemic restrictions are still widely applied in many countries or have been relaxed in the very recent past (after April 2021 in France for example). I am not fully aware of the precise situation in Switzerland but quoting the authors stated in their Methods section, restrictions "were eased on March 1st 2021".

Under these circomstances, it appears that this survey may not be investigating the long-term consequences of the primary pandemic restrictions but is rather exploring the "continuous" restrictions that have been applied during the last year. The wording "long-term" could be referring to previous restrictions abandoned for a long time but that still have consequences at the time of writing, which is obviously not the case here. Those consequences are still particularly relevant today, and as much important for the CF community but the terminology emmployed could be rethought.

My other comments are minor:

Abstract : P2, L35-38 : It seems quite unclear whether the following percentages ("37% and 33% reported to undertake home-based training and 37 outdoor activities more frequently; 22% reported an increase in routine PA, and another 8% 38 started new activities.") are related to the whole study sample or precisely the 32 pwCF that increased daily PA. One would implicitely understand the latter, but it should be precised if that is the case.

Introduction: P3, L65-67: In France, the list of public restrictions proposed by the authors should have encompassed the closing of physiotherapy practices. Should it have also been the case in Switzerland, it should be noticed since many CF patients were advised to continue their physiotherapy treatments (including exercise training) at home, most of them without any supervision.

Methods: P4, L92-101: Did the authors calculated the number of potential responders for this survey ? Considering the distribution channels employed, it appears that a large amount of pwCF in Switzerland were able to access this questionnaire. This would be helpful to state the representativity of the survey results in the Results and the Limitations section.

Methods: P4, L92: It is stated in the supporting information that the questionnaire was programmed in Redcap. Was Redcap used only for the reporting of each questionnaire to facilitate the statistical analysis ? Or was it used by each of the respondents to report their answers ? If so, it should be stated in the Methods section since it would be a strength of the survey in terms of personal data protection. If not, the platform on which respondents completed the questionnaire should be specified here.

Methods: P5, L96: "on the" is repeated twice

Results: Well reported and well-written section.

Discussion: The discussion section is also well-written. Messages are clear and thoughtful.

Discussion: P14, L241: The authors could precise here the percentage of people fully vaccinated, or partially vaccinated in the general population in Switzerland. Vaccination coverage is still very heterogeneous between European countries today and it would be interesting to add a comparative element to support the results.

6. PLOS authors have the option to publish the peer review history of their article (what does this mean?). If published, this will include your full peer review and any attached files.

Reviewer #1: No

Reviewer #2: No

Reviewer #3: **Yes: **Yann Combret

---

## [Decision Letter · Decision Letter 1]

13 Sep 2021

COVID-19 pandemic restrictions continuously impact on physical activity in adults with cystic fibrosis

PONE-D-21-21163R1

Dear Dr. Radtke,

We’re pleased to inform you that your manuscript has been judged scientifically suitable for publication and will be formally accepted for publication once it meets all outstanding technical requirements.

Kind regards,

Sinan Kardeş, M.D.

Academic Editor

PLOS ONE

Reviewers' comments:

Reviewer's Responses to Questions

**Comments to the Author**

1. If the authors have adequately addressed your comments raised in a previous round of review and you feel that this manuscript is now acceptable for publication, you may indicate that here to bypass the “Comments to the Author” section, enter your conflict of interest statement in the “Confidential to Editor” section, and submit your "Accept" recommendation.

Reviewer #1: All comments have been addressed

Reviewer #2: All comments have been addressed

Reviewer #3: All comments have been addressed

2. Is the manuscript technically sound, and do the data support the conclusions?

Reviewer #1: Yes

Reviewer #2: Yes

Reviewer #3: Yes

3. Has the statistical analysis been performed appropriately and rigorously? 

Reviewer #1: Yes

Reviewer #2: Yes

Reviewer #3: Yes

4. Have the authors made all data underlying the findings in their manuscript fully available?

Reviewer #1: (No Response)

Reviewer #2: Yes

Reviewer #3: Yes

5. Is the manuscript presented in an intelligible fashion and written in standard English?

Reviewer #1: Yes

Reviewer #2: Yes

Reviewer #3: Yes

6. Review Comments to the Author

Reviewer #1: (No Response)

Reviewer #2: (No Response)

Reviewer #3: To my opinion, all the changes made on the primary version of this manuscript addressed both my comments and those of the other reviewers. The manuscript could now be accepted in its current form.

7. PLOS authors have the option to publish the peer review history of their article (what does this mean?). If published, this will include your full peer review and any attached files.

Reviewer #1: No

Reviewer #2: No

Reviewer #3: **Yes: **Yann Combret

---

## [Editor Report · Acceptance letter]

16 Sep 2021

PONE-D-21-21163R1 

COVID-19 pandemic restrictions continuously impact on physical activity in adults with cystic fibrosis 

Dear Dr. Radtke:

I'm pleased to inform you that your manuscript has been deemed suitable for publication in PLOS ONE. Congratulations! Your manuscript is now with our production department. 

Kind regards, 

on behalf of

Dr. Sinan Kardeş 

Academic Editor

PLOS ONE